# Characterization of Extraintestinal Pathogenic *Escherichia coli* Strains Causing Canine Pneumonia in China: Antibiotic Resistance, Virulence Genes, and Sequence Typing

**DOI:** 10.3390/vetsci11100491

**Published:** 2024-10-10

**Authors:** Jianyi Lai, Haibin Long, Zhihong Zhao, Gan Rao, Zhaojia Ou, Jiajie Li, Zhidong Zhou, Minhua Hu, Qingchun Ni

**Affiliations:** Guangzhou General Pharmaceutical Research Institute Co., Ltd., Guangzhou 510240, China; 13418109262@163.com (J.L.); happen21@163.com (H.L.); zzh7797896@126.com (Z.Z.); raogan@gpri.com.cn (G.R.); ouu1323792228@163.com (Z.O.); 13030276252@163.com (J.L.); zhouzhidong@gpri.com.cn (Z.Z.)

**Keywords:** puppies, ExPEC, identification, antimicrobial susceptibility test, virulence genes, ST131

## Abstract

**Simple Summary:**

The increase in respiratory infections, particularly acute pneumonia, in puppies poses a significant health risk due to the disease’s rapid severity. This study successfully identified four multidrug-resistant strains of extraintestinal pathogenic *Escherichia coli* from puppies with acute pneumonia, which contained various virulence genes and were classified using sequence typing. This study emphasizes the need to understand the molecular traits of *Escherichia coli* strains causing canine pneumonia. Such an understanding facilitates the formulation of targeted intervention strategies and the exploration of novel therapeutic approaches for the effective treatment of acute pneumonia in puppies.

**Abstract:**

To determine the etiological agents responsible for acute pneumonia in puppies in China, this study utilized bronchoalveolar lavage (BAL) fluid extraction to enable the isolation, culture, biochemical identification, and 16S rRNA PCR amplification of the pathogens. Following preliminary identification, the pathogens underwent analysis for antibiotic resistance phenotypes and resistance genes. Additionally, the study examined the presence of virulence genes, conducted multilocus sequence typing (MLST), and performed whole-genome sequencing (WGS). The findings revealed that all four isolated pathogens were characterized as extraintestinal pathogenic *Escherichia coli* (ExPEC). The examined ExPEC strains demonstrated resistance to cephalosporins, tetracyclines, and penicillins, while remaining susceptible to aminoglycosides, beta-lactamase inhibitors, carbapenems, chloramphenicols, and sulfonamides. An analysis of virulence genes identified the presence of eight genes, namely *CNF-I*, *fyuA*, *fimC*, *papC*, *ompA*, *fimH*, *irp2*, and *iroN*, which are implicated in their invasiveness and potential to inflict tissue damage. The MLST analysis revealed that all ExPEC strains were classified under either sequence type ST131 (Achtman database) or ST43 (Pasteur database). The study further determined that these strains were absent in the kennel’s drinking water source, thereby ruling out water contamination as a potential factor in the emergence of ST131-type ExPEC. This study offers a theoretical framework and empirical evidence for elucidating the potential pathogenic mechanisms and clinical therapeutic strategies of ExPEC in the etiology of acute pneumonia in puppies.

## 1. Introduction

In recent years, there has been a notable rise in the incidence of respiratory infections among canines, with certain cases exhibiting contagious properties that significantly compromise the health and quality of life of affected dogs. This trend also poses substantial challenges to the daily management and breeding operations within kennels. ExPEC, a prevalent Gram-negative bacterium in animals, has been identified as a causative agent of various infections, including respiratory infections, urinary tract infections, and bacteremia, among others [1]. Several studies have isolated and identified ExPECs in the lungs of animals. For instance, Edward B. et al. isolated pathogenic ExPEC from a 7-month-old dog presenting with acute necrotizing pneumonia [2]. A U.S. study conducted between 2013 and 2021 confirmed ExPEC in all 21 cases of canine hemorrhagic pneumonia through pathogen cultivation and identification [3]. Additionally, Yu Y. et al. also identified 85 strains of *Escherichia coli* from 115 mink lung samples affected by hemorrhagic pneumonia [4]. All animals in these studies exhibited symptoms such as lethargy, anorexia, and respiratory distress. Necropsies revealed varying degrees of lung hemorrhages, with hemorrhagic pneumonia identified as the primary cause of death in all cases. These findings bear a striking resemblance to the symptoms observed in four recent sporadic cases of acute pneumonia in puppies at a kennel in Guangdong Province, China.

The classical clinical diagnosis and identification of ExPEC in animals primarily rely on the investigation of medical history, clinical symptomatology, biochemical identification, and antimicrobial susceptibility testing. This approach allows treatment with appropriate drugs, typically obviating the need for further differentiation [5,6]. However, with the gradual increase in the number of ExPEC cases, traditional methods are insufficient to fully elucidate the epidemiology, pathogenesis, and phylogenetic characteristics of ExPEC infections. Moreover, despite the availability of analogous cases for reference, the determination of whether the acute pneumonia observed in puppies at a kennel in Guangdong Province is indeed attributable to ExPEC, and whether it constitutes a broader threat to the canine population remains an urgent issue requiring further investigation. Therefore, this study identified pathogens in puppies with acute pneumonia using microbiological tests and antimicrobial susceptibility testing, followed by bacterial virulence gene detection, MLST, and WGS. These methodologies facilitated the identification of previously unrecognized pathogens of this disease, thereby elucidating their pathogenic mechanisms and informing strategies for future prevention and therapeutic interventions.

## 2. Materials and Methods

### 2.1. Main Reagents and Main Instruments

TSA, TSB, blood agar plates, MacConkey agar, eosin methylene blue agar, Gram staining solutions, Escherichia coli biochemical identification kits, and McFarland standard turbidity tubes were all procured from Guangdong HuanKai Microbial Sci. & Tech. Co., Ltd., Guangzhou, China. Bacterial genomic DNA extraction kits were obtained from Tiangen Bio-chemical Technology Co., Ltd., Beijing, China Green Taq Mix was sourced from Nanjing Novozan Biotech Co., Ltd., Nanjing, China. Gel Red was acquired from Shanghai Biyuntian Biotechnology Co., Ltd., Shanghai, China. The DL-2000 DNA Marker was procured from TaKaRa, Osaka, Japan. Antibiotic susceptibility test disks were purchased from Hangzhou Microbial Reagent Co., Ltd., Hangzhou, China. The ultra-clean workbench was procured from Suzhou Purification Equipment Co., Ltd., Suzhou, China. The high-pressure autoclave was sourced from Shanghai Boxun Medical Biological Instrument Co., Ltd., Shanghai, China, while the electric thermostatic incubator was obtained from Shanghai Yiheng Scientific Instrument Co., Ltd., Shanghai, China. The desktop thermostatic shaker was acquired from Shanghai Hecheng Instrument Manufacturing Co., Ltd., Shanghai, China, and the research-grade upright microscope was supplied by Olympus Corporation, Tokyo, Japan. Additionally, the high-speed centrifuge was provided by Hunan Xiangyi Laboratory Instrument Development Co., Ltd., Hunan, China. The PCR thermal cycler was purchased from Biometra, Germany. The universal electrophoresis apparatus was obtained from Junyi Dongfang Electrophoresis Equipment Co., Ltd., Beijing, China, and the multifunctional chemiluminescence imaging system was acquired from UVP, CA, USA.

### 2.2. Animals

The experimental samples were aseptically collected from four puppies (one male and three females) aged between 3 and 5 months and weighing an average of 6.7 ± 1.4 kg, all of which were diagnosed with acute pneumonia and housed in a kennel in Guangdong Province.

### 2.3. Experimental Methods

#### 2.3.1. Case History Investigation

In a kennel in Guangdong Province, four sporadic instances of acute pneumonia in puppies were documented between December 2023 and May 2024. The clinical manifestations predominantly encompassed lethargy, anorexia, tachypnea with pronounced abdominal effort, and hypersalivation. Following the identification of the cases, the broad-spectrum antibacterial agent ampicillin sodium (approval number: 070011304) was promptly administered intravenously as an emergency intervention at a dosage of 30 mg/kg, administered twice daily for two consecutive days. Despite these therapeutic interventions, the conditions of the dogs continued to deteriorate rapidly. Some cases exhibited the exudation of a reddish foamy substance from the nasal and oral cavities. All cases showed a rapid onset, severe clinical presentation, and poor prognosis, ultimately resulting in mortality within 72 h of initiating treatment. Furthermore, the investigation revealed that all the puppies that exhibited illness had received vaccinations for canine distemper, adenovirus, parainfluenza, parvovirus, coronavirus, and leptospirosis serotypes, attaining standard immunization levels. This finding excluded the likelihood of prevalent common severe infectious diseases in the canine population.

#### 2.3.2. Isolation, Cultivation, and Identification of Pathogens

In an aseptic surgical procedure, puppies afflicted with severe pneumonia were anesthetized. Subsequently, three BAL fluid samples, each with a volume of 2–3 mL, were collected from each canine subject. These samples were then injected into Amies transport medium containing activated charcoal and transported to Zoetis Inc. at ambient temperature. Upon arrival, the samples underwent bacterial isolation, culture, purification, and biochemical identification. Universal primers for bacterial 16S rRNA were designed following the methodology described by I Dewa Made Sukrama et al. [7] for PCR amplification and bidirectional sequencing of the 16S rRNA fragments. The primer sequences are detailed in Appendix B, Table A1. Following the extraction of DNA from four purified bacterial strains, it was extracted using a bacterial genome extraction kit. PCR was performed in a 25 μL reaction mixture containing 12.5 μL of Premix Ex Taq, 2 μL of DNA, 8.5 μL of ddH_2_O, and 1 μL each of forward and reverse primers. The amplification protocol included initial denaturation at 95 °C for 3 min; 35 cycles of 95 °C for 15 s, 57.5 °C for 15 s, and 72 °C for 1 min; and a final extension at 72 °C for 5 min. PCR products were analyzed on a 1% agarose gel at 150 V and 500 mA for 30 min. Upon observing a single band without non-specific amplification, 10 μL of the unpurified PCR products were submitted to Sangon Biotech Co., Ltd., (Shanghai, China) for sequencing.

#### 2.3.3. Pathological Autopsy Observations

After euthanizing the puppies, the thoracic cavity was opened to examine lung morphology and pathology. Typical lung tissue lesions were fixed in 10% neutral buffered formalin, processed for routine paraffin embedding, sectioned, and stained with hematoxylin and eosin (H&E) for a microscopic histological analysis.

#### 2.3.4. Bacterial Whole-Genome Sequencing

The DNA of the four purified strains was extracted and dispatched to Sangon Biotech Co., Ltd., (Shanghai, China) for high-throughput sequencing utilizing the Illumina platform. The raw data (RawReads) underwent rigorous quality assessment and quality control (QC) procedures, which included the removal of adapter sequences, low-quality sequences, and contaminants, resulting in clean data (clean reads). The genome assembly was performed using software tools such as SPAdes (v. 3.5.0) and Pilon (v. 1.23) to generate contigs and scaffolds. Gene prediction and annotation in the assembled genome were conducted using software like Prokka (v. 1.10) to identify coding sequences (CDSs), RNA genes, and other genomic features. Drug resistance genes and virulence factors were identified utilizing databases including the Comprehensive Antibiotic Resistance Database (CARD) and the Virulence Factors Database (VFDB). The MLST analysis was conducted using the PubMLST database to ascertain the sequence type of the strain. Subsequently, SerotypeFinder (v. 2.0) software was employed to conduct serotyping by analyzing sequences of specific O antigen genes (*wzx*, *wzy*, *wzm*, and *wzt*) and H antigen genes (*fliC*, *flkA*, *fllA*, *flmA*, and *flnA*).

#### 2.3.5. Bacterial Antimicrobial Phenotype and Resistance Genes

Typical colonies from the purified bacterial strains were selected and suspended in saline to achieve a turbidity equivalent to a 0.5 McFarland standard turbidity tube (corresponding to a bacterial concentration of 1 × 10^8^ CFU/m L). The bacterial suspensions were subsequently adjusted to a concentration of 1 × 10^8^ CFU/m L. A volume of 100 μL of the adjusted bacterial suspension was evenly spread onto TSA plates (using a sterile glass rod. Different antibiotic disks were then placed on the agar using sterile tweezers, ensuring a distance of 1.5 cm to 2 cm between each disk. Following a 15 min settling period, the plates were incubated in an inverted position overnight at 37 °C. Subsequent observations involved measuring the diameters of the inhibition zones using a caliper. According to the Clinical and Laboratory Standards Institute (CLSI) document VET01S (CLSI, 2024) [8], the test standards delineated in Table 1 were employed to document the drug resistance phenotype of the strain. Additionally, the resistance genes of the bacteria were analyzed based on the WGS results.

#### 2.3.6. Virulence Gene Detection

Utilizing purified strain DNA as a template, the detection of virulence genes in four isolated pathogens was performed using PCR. In accordance with references [4,9,10], primers were designed for 18 types of virulence genes spanning the following six categories: genes related to invasiveness (e.g., *hlyF*), iron uptake (e.g., *fyuA*, *iucD*, *irp2*, *iroN*), adhesion (e.g., *fimC*, *papC*, *fimH*, *tsh*, *K99*), toxin production (e.g., *CNF-I*, *vat*, *STa*, *STb*), biofilm formation (e.g., *ompA*), serum resistance (e.g., *cva/cvi*, *iss*), and the LEE pathogenicity island (e.g., *eae*). The sequences of these primers are provided in Appendix B, Table A1. The amplification process is performed in a 25 μL reaction system with an initial denaturation at 95 °C for 3 min, followed by 35 cycles at 95 °C for 30 s, 55.3 °C for 30 s, and 72 °C for 1 min, ending with a final extension at 72 °C for 5 min. PCR products are analyzed using 1% agarose gel electrophoresis.

#### 2.3.7. MLST

Utilizing the PubMLST database (URL http://pubmlst.org, accessed on 6 June 2024) in conjunction with the relevant literature [11], primer sequences were designed for seven housekeeping genes of *Escherichia coli*, namely *adK*, *fumC*, *gyrB*, *icd*, *mdh*, *purA*, and *recA*. The specific primer sequences are provided in Appendix B, Table A1. Purified strain DNA served as the template for the PCR amplification of these seven housekeeping genes. The resulting PCR products were subsequently submitted to Sangon Biotech Co. Ltd. (Shanghai, China) for bidirectional sequencing. The sequencing results were submitted electronically to the PubMLST Achtman database for comparative analysis. The allele numbers for each of the seven housekeeping gene loci were retrieved. Upon comparing the seven genes of each strain, an allelic profile comprising seven numeric alleles was established. Entering this allelic profile into the MLST database yielded a sequence type (ST), thus determining the multilocus sequence type of the strain.

#### 2.3.8. Detection of *Escherichia coli* and CNF-I Virulence Gene in Drinking Water Sources

A total of 19 tap water samples from a kennel were collected. A total of 1 mL of each sample was dispensed onto TSA solid culture medium using the pour plate method and incubated at 37 °C for 48 h. Upon the development of visible colonies, colonies for further culture in TSB medium overnight were selected. Subsequently, DNA was extracted from the cultured broth. PCR was conducted using universal primers for *Escherichia coli* (primer sequences are provided in the appendix) and primers specific to the *CNF-I* virulence gene. Following PCR amplification, an appropriate volume of the PCR product was subjected to 1% agarose gel electrophoresis for analysis.

## 3. Results

### 3.1. Identification of the Pathogen

Upon inoculation on blood agar plates, the pathogen exhibited non-hemolytic, gray-white large colonies. On MacConkey agar, the colonies appeared red, while on eosin methylene blue agar, they displayed a green metallic sheen with a black center. Gram staining of the suspected colonies confirmed the presence of typical red Gram-negative bacilli. Biochemical identification assays demonstrated that the pathogen was capable of metabolizing glucose, lactose, and sucrose, resulting in the production of acid and gas. However, it was unable to ferment H_2_S, KCN, or urea. The organism demonstrated the ability to degrade tryptophan in peptone water, resulting in the production of indole, which aligns with the characteristic biochemical reactions of *Escherichia coli*. PCR amplification of 16S rRNA was conducted on four purified strains, each producing a target band of approximately 1500 base pairs. Bidirectional sequencing results, analyzed using SnapGene and compared with established *Escherichia coli* sequences in the NCBI database, revealed a 100% homology. These findings conclusively preliminarily identified the isolated pathogens as *Escherichia coli*.

### 3.2. Pulmonary Histology

Upon dissection of the thoracic cavity, extensive hemorrhaging was observed in the lungs (Figure 1a), with the entire lung exhibiting a dark red coloration. In another canine specimen, scattered gray-white necrotic lesions were evident (Figure 1b), accompanied by a malodorous scent. The histopathological examination revealed significant destruction of the fundamental lung architecture, marked proliferation of the interstitial tissue, and infiltration by inflammatory cells, with localized alveolar hemorrhage (Figure 2). These findings suggest that the infection predominantly affects the lower respiratory tract.

### 3.3. Evaluation of Whole-Genome Sequencing Data Quality for Bacteria

For the four purified bacterial strains, a total of 40,443,682 raw sequencing reads (raw reads) were obtained. The number of high-quality sequences (clean reads) ranged from 7,878,437 to 12,421,978 across the samples. The Q30 value for each sample exceeded 93%, and the GC content was at least 50%. These metrics indicate that the sequencing results are of high quality and suitable for subsequent analysis and research (Table 2).

### 3.4. Antibiotic Susceptibility Testing

The antibiotic susceptibility of four isolated strains was evaluated against 21 antimicrobial agents spanning nine categories using the Kirby–Bauer agar diffusion method, with the results detailed in Table 3. The four strains demonstrated sensitivity to amino glycosides, β-lactamase inhibitors, carbapenems, chloramphenicols, and sulfonamides, except for gentamicin. Conversely, they exhibited intermediate resistance and complete resistance to 10 antibiotics, including cephalosporins, tetracyclines, and penicillins. The WGS analysis demonstrated a 100% detection rate for the β-lactamase gene families *TEM* and *CTX-M-55*, the methicillin resistance-associated gene *mec*, the tetracycline resistance gene family *tet*, the aminoglycoside acetyltransferase gene *AAC-IId*, the chloramphenicol acetyltransferase gene *cmlA*, the macrolide resistance-associated gene *dfrA*, and the sulfonamide resistance-associated gene *sul*, suggesting that the isolates harbor multiple resistance genes. Notably, no *SHV* or *qnr* genes were identified. The primary drug resistance genes identified in the four strains are enumerated in Table 4. The classifications of gene types and their corresponding mechanisms of drug resistance have been sourced from the CARD (https://card.mcmaster.ca/home, accessed on 25 August 2024). By synthesizing these findings according to the CLSI document VET01S (CLSI, 2024) [8], all four isolates were identified as *Escherichia coli* that produce extended-spectrum β-lactamases (ESBLs).

### 3.5. Detection of Virulence Genes

In this experiment, the presence of virulence genes in the four isolated bacterial strains was assessed using PCR. The analysis revealed the detection of eight specific genes, namely *CNF-I*, *fyuA*, *fimC*, *papC*, *ompA*, *fimH*, *irp2*, and *iroN*. Additionally, the core virulence genes present in the four strains through WGS are comprehensively detailed in Table 5. The classification and functional analysis of these genes were obtained from the VFDB (http://www.mgc.ac.cn/VFs, accessed on 25 August 2024).

### 3.6. MLST

The PCR amplification of seven housekeeping genes was conducted on the four isolated strains, yielding fragments of the anticipated sizes. The bidirectional sequencing data were consolidated using SnapGene software (v. 6.0.2). Based on the sequencing results of the seven *Escherichia coli* housekeeping genes, allele numbers were assigned in accordance with the PubMLST Achtman database as follows: *adk*—53; *fumC*—40; *gyrB*—47; *icd*—13; *mdh*—36; *purA*—28; and *recA*—29. Consequently, the ST for all four isolated strains was determined to be ST131. Additionally, the whole-genome sequencing assembly data of the bacterial isolates were submitted to the Pasteur database, which verified that the ST for all four strains was ST43. No new sequence types were identified in this study.

### 3.7. Detection of Escherichia coli and CNF-I Virulence Gene in Drinking Water

The PCR analysis detected the presence of *Escherichia coli* in all tested water samples. However, the *CNF-I* virulence gene was absent in all cases. The ddH_2_O negative control remained uncontaminated. These results indicate that waterborne contamination is unlikely to contribute to the emergence of ST131 ExPEC.

## 4. Discussion

In recent years, the emergence of multidrug-resistant Gram-negative bacteria has elicited significant global concerns within the realms of both human and veterinary medicine. The emergence of these resistant strains presents a significant threat to public health and undermines the efficacy of clinical treatments. In clinical microbiology laboratories, the production of ESBLs in Gram-negative bacteria is characterized by reduced susceptibility to one or more of the following antibiotics: ceftazidime, cefpodoxime, ceftriaxone, cefotaxime, or aztreonam. Furthermore, the antimicrobial efficacy of these antibiotics against ESBL-producing bacteria is augmented in the presence of β-lactamase inhibitors such as clavulanic acid or tazobactam [12]. ESBLs are β-lactamases that can hydrolyze penicillins and cephalosporins, predominantly encompassing the *TEM*, *CTX-M*, *SHV*, and *OXA* families [13,14]. When *Escherichia coli*, a member of the *Enterobacteriaceae* family, acquires ESBL genes and produces these enzymes, it develops resistance to this class of β-lactam antibiotics. This study identified four *Escherichia coli* isolates using 16S rRNA PCR and performed susceptibility testing with nine classes of antimicrobial drugs. The findings indicated a consistent resistance to cephalosporins, tetracyclines, and penicillins, whereas sensitivity was noted towards aminoglycosides, β-lactamase inhibitors, carbapenems, chloramphenicol, and sulfonamides. Furthermore, high-quality sequencing data revealed that all isolates harbored the *TEM*, *CTX-M-55*, *mec*, *tet*, *AAC-IId*, *cmlA*, *dfrA*, and *sul* genes, with *TEM* and *CTX-M-55* being particularly prevalent in companion animal *Escherichia coli* that produces ESBLs [15,16,17]. The *TEM-206* variant identified in this study represents a unique beta-lactamase, typically categorized as a specific third-generation ESBL due to its capability to hydrolyze a broad range of beta-lactam antibiotics, including those classified as third-generation [18]. The presence of this enzyme in bacterial strains imparts resistance to antibiotics commonly employed in the treatment of severe infections. In summary, the observed ESBL production and multidrug resistance of the isolates align with findings from previous studies [19].

The pathogenicity of *Escherichia coli* is predominantly dictated by specific virulence genes, such as invasins, adhesins, toxins, and capsules. These genes are frequently organized into extensive genetic regions located on chromosomes, plasmids, or bacteriophages, and they possess the capability for horizontal transfer between strains [20]. One notable virulence factor is cytotoxic necrotizing factor I (*CNF-I*), a protein toxin synthesized by pathogenic strains. *CNF-I* exerts its effects by permanently activating the regulatory GTPases Rho, Rac, and Cdc42 in eukaryotic cells through the deamidation of glutamine residues. This activation promotes gene transcription and cell proliferation, thereby enhancing bacterial survival [21]. A 2021 study investigating hemorrhagic pneumonia in a range of animal species, including canines, felines, tigers, and equines, analyzed the virulence genes in *Escherichia coli* isolates. The findings revealed that all isolates tested positive for *CNF-I* [22]. Furthermore, numerous studies have demonstrated that *CNF-I* is crucial in the pathogenesis of *Escherichia coli*-induced hemorrhagic pneumonia [3,4,23]. Given that the presence of a single virulence gene is insufficient to convert an *Escherichia coli* strain into a pathogenic form and instead requires the interaction of multiple virulence genes [14], this study aimed to detect 18 specific virulence genes in the isolates. The results revealed that the bacteria harbored *CNF-I*, *fyuA*, *fimC*, *papC*, *ompA*, *fimH*, *irp2*, and *iroN*, while other virulence genes were absent. Notably, the genes *STa*, *STb*, and *eae*, which tested negative in amplification, are associated with intestinal pathogenic *Escherichia coli* (IPEC) and are significant markers for identifying IPEC [20,24]. Consequently, although the identification of ExPEC based solely on virulence genes remains impractical [24], this experiment effectively excludes the isolated strains as IPEC, thereby confirming their classification as ExPEC.

Martin C. initially introduced the MLST technique in 1998. In contrast to conventional molecular typing methods for bacteria, such as random primer PCR and pulsed-field gel electrophoresis, MLST examines the nucleotide sequences within housekeeping genes to identify alleles, thereby facilitating the analysis of strain variations. This technique applies to a broad spectrum of bacterial species, and the results it yields demonstrate high reproducibility across different laboratories [25]. Furthermore, studies have indicated that MLST possesses superior typing capabilities for ESBL-producing *Escherichia coli* compared to pulsed-field gel electrophoresis [26]. As a sequencing technology, WGS offers a more comprehensive view of genomic information, with various databases potentially encompassing different strains and gene sequences [27]. In this study, we employed MLST in conjunction with WGS to analyze four isolated strains. Our results indicated that all strains belonged to ST131 and ST43, as identified by the Achtman and Pasteur databases, respectively, with no novel ST types detected. ST131, a widespread clonal type of *Escherichia coli*, is implicated in a range of human and animal infections, including asymptomatic bacteriuria, upper gastrointestinal infections, septicemia, meningitis, and pneumonia [28,29]. It has consistently been classified as ExPEC in numerous studies [30,31,32]. Other MLST investigations have demonstrated that the predominant STs in clinically isolated *Escherichia coli* from dogs are ST372 and ST73 [30,33,34], whereas the primary STs associated with respiratory infections are ST12 and ST681 [33]. Additionally, the WGS analysis in this experiment revealed that the serotypes of these bacteria were all O25:H4. To date, there have been limited reports on O25:H4/ST131 isolates derived from animal lung infection. Even though the ExPEC responsible for acute pneumonia in four puppies in a kennel in Guangdong Province were all identified as O25:H4/ST131 isolates, there is currently no clinical evidence to suggest significant transmissibility based on data surveys conducted over the past six months. This infection is presently considered incidental, and further research is required to ascertain any additional impacts on the canine population.

ExPEC is a multidrug-resistant bacterium, with 84 cases of acute pneumonia in puppies attributed to it documented in the international literature [2,3,22,23,35,36]; however, no domestic cases have been reported to date. In this study, we employed BAL for the first time to collect pathogenic samples from puppies diagnosed with acute pneumonia. In comparison to traditional methods, such as swabbing lung tissue or performing direct lung tissue excision, BAL offers a minimally invasive alternative that obviates the need for thoracotomy. This technique not only enhances diagnostic safety and efficacy but also significantly promotes animal welfare and facilitates postoperative recovery. Furthermore, these ExPEC studies consistently detected the *CNF-I* virulence gene, aligning with our findings. This underscores the critical role of *CNF-I* in the early diagnosis and intervention of such diseases.

Numerous theories persist regarding the sources and transmission pathways of ST131 ExPEC, encompassing the consumption of animals (particularly poultry), domestic and medical wastewater, interpersonal transmission among veterinary healthcare workers, familial transmission, and occupational contact with animals [37,38,39]. Although this study did not succeed in pinpointing the exact source of infection responsible for sporadic cases in kennels, it was pioneering in the detection of the *CNF-I* gene in drinking water samples. This finding effectively rules out the transmission of ST131 ExPEC through the drinking water source in kennels. However, the study’s scope was limited, as it only examined four cases of acute pneumonia in puppies over six months. Consequently, this is insufficient to definitively determine whether ST131 or ST43 ExPEC is the sole pathogen responsible for causing acute pneumonia in all puppies in the future. Additionally, *Escherichia coli* is a normal colonizing bacterium in the intestine [40]. This raises the question of whether there are analogous colonizing microbial communities in the dog’s lungs. What is the relationship between these pulmonary communities and *Escherichia coli*? The mechanisms underlying the alterations in the lung’s colonizing microbial communities that lead to infection remain unclear, warranting further investigation.

Currently, the antimicrobial agents frequently employed in veterinary clinical practice for the treatment of Gram-negative bacterial infections encompass cephalosporins, aminoglycosides, quinolones, and β-lactamase inhibitors [41]. Before initiating treatment, it is imperative to thoroughly review the dog’s history of antibiotic usage to circumvent the administration of drugs to which resistance may have developed. This study meticulously documented the medication histories of the affected dogs, and informed by the outcomes of drug sensitivity assays, proposed the following treatment recommendations: for canines presenting with mild acute pneumonia, initial therapeutic considerations may include the administration of theophylline or anticholinergic agents to achieve bronchodilation. Concurrently, dexamethasone and furosemide may be employed to mitigate the secretion of pulmonary inflammatory exudate and pleural effusion. Additionally, the utilization of a hyperbaric oxygen chamber for oxygen therapy can be instrumental in maintaining adequate blood oxygen saturation levels, while timely electrolyte replenishment should also be ensured. Antimicrobial therapy may involve the administration of penicillins in conjunction with β-lactamase inhibitors. In severe cases or instances where initial treatment proves ineffective—characterized by blood oxygen saturation levels below 90% and respiratory rates exceeding 30 breaths per minute [42]—it is advisable to integrate considerations of animal welfare and the 3R principle [43]. This includes the timely euthanasia of affected animals, vigilant monitoring of co-housed dogs, and the implementation of appropriate isolation, disinfection, and other preventive measures. Furthermore, to mitigate the progression of drug resistance, it is imperative to minimize the extensive use of broad-spectrum antibiotics. Moreover, reinforcing feeding management practices and augmenting the innate resistance of the canine population are crucial strategies for the prevention and control of this disease.

## 5. Conclusions

This study meticulously investigated the occurrence of sudden and sporadic acute pneumonia in puppies within a kennel in Guangdong Province to elucidate its etiological factors and transmission dynamics. Through a comprehensive analysis of four cases, we successfully isolated and identified four strains of ExPEC that produce ESBLs and exhibit multidrug resistance, as well as a variety of virulence genes. The MLST analysis revealed that the isolated ExPEC strains were classified as either ST131 or ST43. ST131 is not a strain associated with epidemics in canines, nor does it disseminate disease through contaminated drinking water sources. This indicates that acute pneumonia in puppies may not be contagious and could represent an incidental bacterial infection. For effective disease prevention and control, it is crucial to emphasize the judicious use of pharmaceuticals, implement scientifically based breeding management practices, and enhance the immune resilience of the canine population to mitigate the incidence of acute pneumonia in puppies.

The original image of this manuscript could be seen in the Appendix A.

## Figures and Tables

**Figure 1 vetsci-11-00491-f001:**
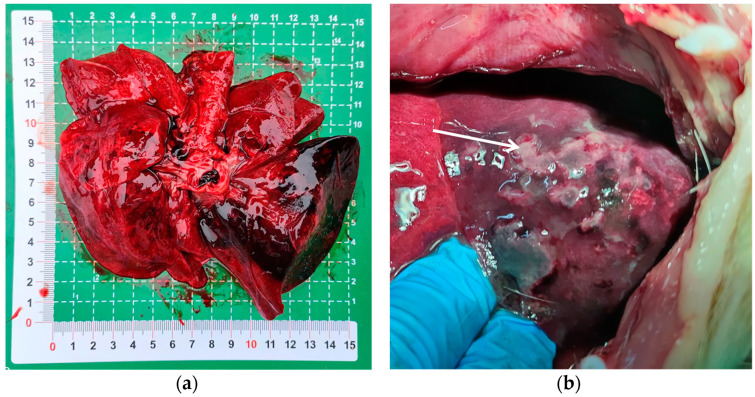
Pulmonary lesions in puppies with acute pneumonia. (**a**) The entire lung displays a dark red coloration; (**b**) the white arrow indicates gray-white necrotic lesions in the lung tissue.

**Figure 2 vetsci-11-00491-f002:**
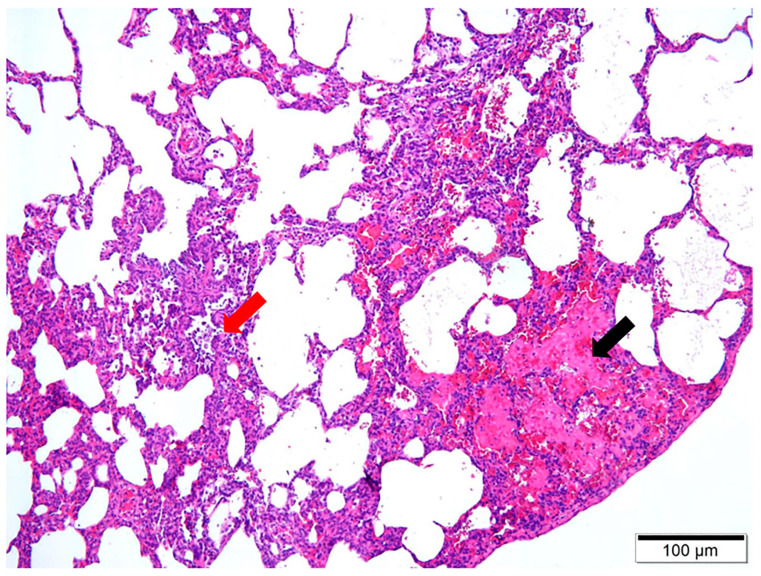
H&E staining of pathological lung tissue at 10× magnification. Two main observations are noteworthy, including localized alveolar hemorrhage, indicated by the black arrow, and the infiltration of inflammatory cells, denoted by the red arrow.

**Table 1 vetsci-11-00491-t001:** Performance standards for antimicrobial susceptibility testing.

Antimicrobial Categories	Antimicrobial Agent	Disk Content/μg	Standards/mm		
			S	I	R
Cephalosporins	Ceftazidime	30	≥21	18–20	≤17
	Cefazolin	30	≥23	20–22	≤19
	Cephalexin	30	≥21	18–20	≤17
	Cefpodoxime	10	≥21	18–20	≤17
	Ceftiofur	30	≥21	18–20	≤17
	Cephalothin	30	≥18	15–17	≤14
Tetracyclines	Tetracycline	30	≥15	12–14	≤11
	Doxycycline	30	≥16	13–15	≤12
Aminoglycosides	Gentamicin	10	≥16	13–15	≤12
	Amikacin	30	≥20	17–19	≤16
	Neomycin	30	≥17	13–16	≤12
Quinolones	Levofloxacin	5	≥21	17–20	≤16
	Pradofloxacin	5	≥24	20–23	≤19
	Marbofloxacin	5	≥20	15–19	≤14
	Enrofloxacin	10	≥19	15–18	≤14
Penicillins	Ampicillin	10	≥17	-	-
β-lactam combination agents	Amoxicillin–clavulanate	20	≥18	–	–
	Piperacillin–tazobactam	100/10	≥25	21–24	≤20
Carbapenems	Imipenem	10	≥23	20–22	≤19
Chloramphenicols	Chloramphenicol	30	≥21	18–20	≤17
Sulfonamides	Cotrimoxazole	23.75/1.25	≥16	10–15	≤9

Abbreviations: S, sensitive; I, intermediary; R, resistant.

**Table 2 vetsci-11-00491-t002:** Quality evaluation of whole-genome sequencing data of bacterial samples.

Sample	Raw Reads	Clean Reads	Average Read Length (bp)	Q30 (%)	GC Content (%)
1	10,005,434	10,000,031	149.10	94.69	50.76
2	10,133,302	10,127,526	149.48	95.60	50.73
3	12,429,312	12,421,978	149.24	96.14	50.58
4	7,883,010	7,878,437	149.79	93.85	50.43

**Table 3 vetsci-11-00491-t003:** Antibiotic resistance phenotypes.

Antimicrobial Categories	Antimicrobial Name	Antibiotic Susceptibility Testing			Sensitivity
		R	I	S	
Cephalosporins	Ceftazidime	3	1	0	0%
	Cefazolin	4	0	0	0%
	Cephalexin	4	0	0	0%
	Cefpodoxime	4	0	0	0%
	Ceftiofur	4	0	0	0%
	Cephalothin	4	0	0	0%
Tetracyclines	Tetracycline	4	0	0	0%
	Doxycycline	1	3	0	0%
Aminoglycosides	Gentamicin	1	0	3	75%
	Amikacin	0	0	4	100%
	Neomycin	0	0	4	100%
Quinolones	Levofloxacin	1	0	3	75%
	Pradofloxacin	0	0	4	100%
	Marbofloxacin	0	0	4	100%
	Enrofloxacin	0	0	0	100%
Penicillins	Ampicillin	4	0	0	0%
β-lactamase inhibitors	Amoxicillin-clavulanate	0	0	4	100%
	Piperacillin-tazobactam	0	0	4	100%
Carbapenems	Imipenem	0	0	4	100%
Chloramphenicol	Chloramphenicol	0	0	4	100%
Sulfonamides	Cotrimoxazole	0	0	4	100%

Abbreviations: S, sensitive; I, intermediary; R, resistant.

**Table 4 vetsci-11-00491-t004:** Presence of genes related to antimicrobial resistance.

Gene Name	Gene Type	Mechanism
*TEM-206*	TEM beta-lactamase	antibiotic inactivation
*CTX-M-55*	CTX-M beta-lactamase	
*mecA*	methicillin-resistant PBP2	antibiotic target replacement
*mecC*		
*tet(X4)*	tetracycline inactivation enzyme	antibiotic inactivation
*tetB(P)*	tetracycline-resistant ribosomal protection protein	antibiotic target protection
*tet(Q)*		
*tetB(60)*	ATP-binding cassette (ABC) antibiotic efflux pump	antibiotic efflux
*tetA(46)*		
*tetB(46)*		
*tetA(60)*		
*tetB(46)*		
*tet(30)*	major facilitator superfamily (MFS) antibiotic efflux pump	
*tet(C)*		
*tetA(58)*		
*cmlA9*		
*cmlA6*		
*AAC(3)-IId*	AAC(3)	antibiotic inactivation
*dfrA21*	trimethoprim-resistant dihydrofolate reductase dfr	antibiotic target replacement
*sul2*	sulfonamide resistant sul	
*sul4*		

**Table 5 vetsci-11-00491-t005:** Presence of genes related to virulence.

Gene Name	Gene Type	Function
*CNF-I*	Cytotoxic necrotizing factor 1	Induces the formation of actin stress fibers and membrane ruffling, necrosis
*fim2*	Serotype 2 fimbrial subunit precursor	Fimbriae may mediate the binding of Bordetella to respiratory epithelium via the major fimbrial subunits and to monocytes via *FimD*
*fimA*	Type 1 fimbrial protein, A chain precursor, fimbrial protein	Makes an important contribution to the colonization of the bladder
*fimB*	Type 1 fimbriae regulatory protein fimB	
*fimD*	Outer membrane usher protein *fimD* precursor	
*fimE*	Type 1 fimbriae regulatory protein *fimE*	
*fimF*	FimF protein precursor	
*fimG*	FimG protein precursor	
*fimH*	FimH protein precursor	
*fimI*	Fimbrin-like protein fimI precursor	
*fimC*	Chaperone protein *fimC* (precursor)	Adherence; invasion
*fyuA*	Pesticin/yersiniabactin receptor	*FyuA/Psn-Irp* system uses yersiniabactin, a siderophore that can remove iron from a number of mammalian proteins due to its extremely high affinity for ferric iron
*irp1*	Yersiniabactin biosynthetic protein Irp1	
*Irp2*	Yersiniabactin biosynthetic protein Irp2	
*iroN*	Salmochelin receptor IroN	Catecholate siderophore receptor, mediates utilization of the siderophore enterobactin
*ompA*	Outer membrane protein P5 (*ompA*), human factor H binding protein	Interacts with CEACAM1, a member of the carcinoembryonic antigen (CEA) family of cell adhesion molecules, a glycoprotein expressed by respiratory epithelial cell
*papA*	P pilus major subunit PapA	*Pap* pili expression, which is mannose-resistant, is tightly regulated by environmental and nutritional factors and a methylation-dependent phase variation mechanism. The pap operon is a key example of pilus assembly, featuring conserved elements like *PapD*, an Ig-like domain chaperone essential for transporting pilus subunits from the cytoplasm to the outer membrane. *PapD*–subunit complexes are directed to the *PapC* usher in the outer membrane, forming a pore for pilus translocation. The main subunit, *PapA*, assembles into a 6.8 nm helical rod anchored by *PapH*. The pilus rod’s distal end has a 2 nm linear tip fibrillum made of *PapE* connected to the *PapA* rod by *PapK*. *PapG* attaches to the PapE tip fibrillum via the adapter protein *PapF*.
*papB*	Regulatory protein PapB	
*papC*	Usher protein PapC	
*papE*	P pilus minor subunit PapE	
*papF*	P pilus minor subunit PapF	
*papG*	P pilus tip adhesin PapG	
*papH*	P pilus termination subunit PapH	
*papI*	Regulatory protein PapI	
*papJ*	P pilus assembly protein PapJ	
*papK*	P pilus minor subunit PapK	

## Data Availability

Data are available upon reasonable request.

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
