# Peer review of "Characterization of Extraintestinal Pathogenic Escherichia coli Strains Causing Canine Pneumonia in China: Antibiotic Resistance, Virulence Genes, and Sequence Typing"

_vetsci, 2024, doi:10.3390/vetsci11100491_

Round 1

Reviewer 1 Report

Comments and Suggestions for Authors

This paper characterized the pathogenicity, antimicrobial resistance, and sequence types of extraintestinal pathogenic E. coli isolates as a causative agent of pneumonia in puppies. Although the paper has a certain novelty, it needs to be more clearly differentiated from previous reports. The following points should be considered.

Major revision

Differences from previous reports

As described by the authors, there have been several reports of E. coli as a cause of pneumonia in dogs. Although the present paper is the first report of E. coli in Guangzhou, China, the authors have not sufficiently explained the differences from previous reports, except for the geographical factor. The authors need to fully explain the novelty that does not exist in the other similar investigations.

Treatment History

 The treatment history of the pups in this paper was not mentioned. The authors should describe treatment history in detail (e.g., what treatment was given?, how the pups progressed?). This would be very informative to the readers and should be included.

Alveolar lavage fluid findings

 The authors isolated E. coli from alveolar lavage fluid, but the cytology results are important to confirm that these E. coli are causative organisms (e.g., Are neutrophils detected?). The authors should describe the cytological results of alveolar lavage fluid.

Description of the WGS technique

 Although the authors carried out WGS, they did not explain the method in detail. The authors should provide a detailed description of the WGS methodology. Apart from that, drug resistance genes, virulence genes, and MLST analysis were conducted by ordinary PCR, but their significance is lacking because all of them can be analyzed by WGS. The authors should explain the significance of conducting conventional PCR and sequencing.

Analysis of ESBL-producing E. coli

 Further analysis of ST131 CTX-M-15-producing E. coli is needed based on the results of WGS. In particular, the analysis results of serotypes and fimbrie types are essential. In addition, the results of the virulence genes, antimicrobial resistance genes, in addition to serotypes and fimbrie types, detected by WGS should be newly listed in a table.

Revision of Figures

 Figures 1, 2, 3, 6, 7, and 8 are unnecessary in this paper and should be deleted.

Minor revision

L241: TEM should be typed based on sequencing because this-type beta-lactamase contains non-ESBL and ESBL.

L248: NCCLS guideline does not exist at present, and the authors should check the latest guideline.

Table 2: Inhibition zone diameter is not necessary and should be deleted. Only phenotype results should be presented. 

Reviewer 2 Report

Comments and Suggestions for Authors

The manuscript entitled "Characterization of Extraintestinal Pathogenic Escherichia coli Strains Causing Canine Pneumonia in China: Antibiotic Resistance, Virulence Genes, and Sequence Typing” was evaluated. Researchers reported that four E.coli strains had been isolated from dogs with acute pneumonia, antibiotic resistance phenotypes, sistance genes, virulence genes and multilocus sequence typing had been determined. This manuscript helps to understand of the etiology of acute pneumonia in puppies. The paper is in the scope of the journal and may be published.

Negative aspects

Line 65-69: this sentence should corrected for clarity.

3.1. Identification of the Pathogen: how many clones were identified in each plate? And whether all clones were E.coli?

3.4. Antibiotic Susceptibility Testing: the standards of the tests should be included.

Line 333-359: The four strains were belonged to ST131 by MLST, and were belonged to ST43 by WGS. Why these strains could be divided in two ST types?

Round 2

Reviewer 1 Report

Comments and Suggestions for Authors

This paper was improved by the author's revisions.

You should clearly describe the drugs administered to puppies (Line 263-265).
